# Synergic Effect of Metformin and Everolimus on Mitochondrial Dynamics of Renal Cell Carcinoma

**DOI:** 10.3390/genes13071211

**Published:** 2022-07-06

**Authors:** Seong-Hwi Hong, Kwang-Suk Lee, Hyun-Ji Hwang, Sung-Yul Park, Woong-Kyu Han, Young-Eun Yoon

**Affiliations:** 1Department of Urology, Hanyang University College of Medicine, Seoul 04763, Korea; hshshshsh90@hanyang.ac.kr (S.-H.H.); hjamelia@hanyang.ac.kr (H.-J.H.); syparkuro@hanyang.ac.kr (S.-Y.P.); 2Department of Urology, Yonsei University College of Medicine, Seoul 03722, Korea; calmenow@yuhs.ac; 3Department of Translational Medicine, Hanyang University Graduate School, Seoul 04763, Korea; 4Department of Medical and Digital Engineering, Hanyang University Graduate School, Seoul 04763, Korea

**Keywords:** everolimus, metformin, mitochondrial dynamics, renal cell carcinoma

## Abstract

Renal cell carcinoma (RCC) frequently recurs or metastasizes after surgical resection. Everolimus, an mTOR inhibitor, is used as a second-line treatment, but the response of RCC to everolimus is insufficient. Metformin is an antidiabetic drug; recent reports have indicated its anti-cancer effects in various cancers, and it is known to have synergistic effects with other drugs. We investigated the possibility of coadministering everolimus and metformin as an effective treatment for RCC. RCC cells treated with a combination of the two drugs showed significantly inhibited cell viability, cell migration, and invasion, and increased apoptosis compared to those treated with each drug alone. An anti-cancer synergistic effect was also confirmed in the xenograft model. Transcriptome analysis for identifying the underlying mechanism of the combined treatment showed the downregulation of mitochondrial fusion genes and upregulation of mitochondrial fission genes by the combination treatment. Changes in mitochondrial dynamics following the combination treatment were observed using LysoTracker, LysoSensor, and JC-1 staining. In conclusion, the combination of everolimus and metformin inhibited RCC growth by disrupting mitochondrial dynamics. Therefore, we suggest that a treatment combining metformin and everolimus disrupts mitochondrial dynamics in RCC, and may be a novel strategy for RCC treatment.

## 1. Introduction

Renal cell carcinoma (RCC) is a malignant urological tumor that accounts for 2.2% of all new cancers [1]. Approximately 85% of kidney tumors are RCC, and approximately 70% are diagnosed by clear cell histology [2,3,4]. After surgical excision for localized RCC, 20–30% of patients with a localized tumor experience tumor relapse or metastasis. In the analysis of the Surveillance, Epidemiology, and End Results Program (SEER) database, the 5-year survival of patients with advanced RCC was extremely poor compared to that of patients with localized RCC (7.3–11.7% vs. 88.4–92.6%) [5].

To date, systemic treatment options for metastatic RCC have been limited. Interferon-α and high-dose interleukin-2 were introduced as therapies for metastatic RCC, but are now only used in selected patients [6]. To date, several targeted therapies utilizing tyrosine kinase inhibitors (TKIs) and/or anti-vascular endothelial growth factor (VEGF) antibodies are widely used as first- and second-line treatments. Recently, HIF2a targeting drugs or immune checkpoint inhibitors have been developed, and although their effects are superior to existing drugs [7,8], their effects are still insufficient; combination therapy to increase drug response is therefore being conducted in several studies [9,10,11]. Mammalian target of rapamycin (mTOR) inhibitors, such as temsirolimus and everolimus, are also used in this setting. Several targeted agents have been approved by the FDA [6]. However, their efficacy is insignificant when considering a survival benefit of less than 1 year. mTOR inhibitors are used as second-line treatments, although their survival gain is only 3–5 months [12]. Therefore, improvement of the response to mTOR inhibitors is a requirement.

Metformin is a first-line treatment for type 2 diabetes mellitus (T2DM), and is used as a significant factor in reducing the risk of cancer and cancer-related mortality in T2DM patients [13]. The oncological and survival benefits of metformin have been reported to be dependent on the cancer type. In several studies of diabetic and metastatic RCC patients treated with sunitinib, metformin treatment showed a survival benefit [14,15]. Several experimental and clinical studies have reported the effect of metformin on cell growth in RCC, prostate cancer, breast cancer, hepatocellular cancer, and colorectal cancer [16,17,18,19,20,21,22,23]. Of the several mechanisms of action of metformin in treating cancer, the activation of adenosine 5′-monophosphate-activated protein kinase (AMPK) and the inhibition of mTOR activity are considered the main pathways against RCC [19]. Additionally, metformin reduces glycogenesis, mitogenic effects, and tumor growth in cancer cells under high insulin levels [17,24,25]. In a recent study, metformin, along with everolimus, was reported as a therapeutic option that could affect mitochondrial dysfunction and tumor aggressiveness. [26]

The mechanisms of the combination of metformin and chemotherapy have been reported [27,28,29,30,31,32]. However, the therapeutic effect of combined treatments with metformin and everolimus, as conventional targeted agents of metastatic RCC, requires in-depth research. We investigated whether combination treatment with metformin and everolimus would synergistically enhance the anti-cancer effects in RCC. To test this hypothesis, and to predict the potential clinical value of this combination, we first analyzed the effects of metformin and everolimus on RCC cell lines in vitro; these effects were then validated by in vivo experiments. Additionally, to elucidate the mechanism underlying the anti-cancer effects of the combination of metformin and everolimus, we performed a transcriptome analysis. In summary, we demonstrate that the combination of metformin and everolimus, as a novel therapeutic strategy for RCC, is associated with the imbalance of mitochondrial dynamics.

## 2. Materials and Methods

### 2.1. Cell Lines and Culture Conditions

The human RCC cell lines Caki-1, A498, and ACHN were purchased from the Korean Cell Line Bank (KCLB, Seoul, Korea). Each cell line was maintained in RPMI-1640 or DMEM (Sigma–Aldrich, St. Louis, MO, USA) containing 10% fetal bovine serum (FBS; Sigma–Aldrich) and 1% antibiotic–antimycotic (ThermoFisher Scientific, Waltham, MA, USA). All cells were cultured in an incubator designed to maintain a temperature of 37 °C and high humidity for the growth of tissue culture cells in a 5% CO_2_ atmosphere.

### 2.2. Cell Viability Assay

Cells at 1 × 10^4^ per well were seeded in 96-well plates and incubated overnight in a complete medium. The three RCC cell lines were then treated with various concentrations of everolimus (0, 1, 5, 10, and 25 µM) or metformin (0, 1, 10, 20, and 50 mM) for 24 h. A cell viability assay was performed using an EZ-cytox system (DoGenBio, #EZ-1000, Seoul, Korea) according to the manufacturer’s recommendations. EZ-cytox solution was added at a ratio of 1:10 to 100 μL of culture medium, and the cells were incubated for 1 h in the dark. The absorbance was measured at 450 nm using a microplate reader. The colorimetric values were normalized to the control, and expressed as a percentage of the control. Data are presented as the mean ± standard error of the mean (SEM).

### 2.3. Apoptosis Assay

Apoptotic cells were measured using an annexin V-FITC apoptosis detection kit (#556547; BD Biosciences, Franklin Lakes, NJ, USA). Briefly, the cells were seeded at an equal density of cells in 60-mm cell culture dishes. The following day, the cells were treated with metformin (20 mM) and everolimus (10 μM) for 24 h. Then, the supernatant and trypsinized cells were suspended in 1× annexin V binding buffer to 1 × 10^6^ cells/mL. FITC-conjugated annexin V (5 μL) and PI (2 μL) were added to a suspension of 1 × 10^5^ cells/100 μL. After incubation for 15 min at room temperature in the dark, 400 μL of 1× binding buffer was added to each tube. The cells were analyzed using a FACS Canto flow cytometer (BD Biosciences, San Jose, CA, USA).

### 2.4. Western Blotting

For cell lysis, the cell pellet was suspended in RIPA buffer, incubated on ice for 15 min, and sonicated for 1 min. The cell lysate was boiled with 4× sample buffer, and 30 μg of protein was quantified for each sample. The protein concentrations were measured using a BCA protein assay (ThermoFisher, #23227). Antibodies against phospho-mTOR (Cell Signaling Technology, #2971s, Danvers, MA, USA), mTOR (Cell Signaling Technology, #2972s), phospho-p70S6K (Cell Signaling Technology, #9205s), p70S6K (Cell Signaling Technology, #9202s), phospho-4EBP1 (Cell Signaling Technology, #9459s), 4EBP1 (Cell Signaling Technology, #9452s), and β-actin (GeneTex, #GTX109639, Irvine, CA, USA) were purchased from the indicated companies.

### 2.5. Wound Healing Assay and Invasion Assay

Cells were grown to 80% density, and a thin “wound” was introduced by scraping the cell culture plate with a sterile pipette tip at a constant width. Cells at the wound edge migrated to the wound space. The widths of the initial gap (0 h) and the residual gap, 24 h after wounding, were determined using an optical microscope (Olympus, Shinjuku, Japan). Cell invasion ability was measured using a Boyden chamber-like design (BD Biosciences, San Jose, CA, USA). The top surface of a Transwell chamber, with a pore size of 0.8 μm, was coated with 100 μL of Matrigel (BD Biosciences) diluted to a concentration of 0.3 mg/mL with a coating buffer. After the coating process of incubation at 37 °C for 3 h, the cells were placed on the upper side of the Transwell insert, the insert was placed in a 24-well plate, and 20% FBS as a chemoattractant was added to the lower well. The cells were incubated overnight, and stained with 0.4% crystal violet to identify invading cells. Images of the invading cells were captured using a microscope at 40× magnification.

### 2.6. Gene Expression Microarray

For the microarray analysis, the synthesis process was started with 1500 ng of total RNA using the BeadChip Labeling Kit (EPICENTRE, Madison, WI, USA), and biotinylated cRNA was synthesized. All processes were performed in accordance with the manufacturer’s instructions. Array signals were detected using streptavidin-Cy3 (GE Healthcare Bio-Sciences, Little Chalfont, UK), according to the bead array manual. Chip performance and labeled cRNA quality were monitored with a bead array reader confocal scanner (532 nm laser illumination), in accordance with the manufacturer’s instructions. The raw data for each sample were extracted using the manufacturer-provided software (Illumina GenomeStudio v2011.1, Gene Expression Module v1.9.0) using the default parameters. The array probe was log-transformed, and quantile normalization was applied to normalize the raw data.

### 2.7. TCGA Database Analysis

To determine the mRNA expression levels in RCC, data were downloaded from the TCGA-KIRC project. The cBioPortal for Cancer Genomics website (http://www.cbioportal.org/public-portal/, accessed on 28 January 2016) was used to access the mRNA expression data and genetic mutation ratios in the TCGA-KIRC project.

### 2.8. RNA Extraction and qRT-PCR

Total RNA was extracted using TRIzol reagent, and AccuPower RT Premix (Bioneer, Daejeon, Korea) was used to reverse-transcribe the extracted RNA. For qRT-PCR, cDNA amplification was conducted on a LightCycler^®^ 480II (Roche, Basel, Switzerland) using the LightCycler^®^ 480 SYBR Green I Master Mix (Roche, #04887352001), in accordance with the manufacturer’s recommendations. For qRT-PCR, gene-specific primers were used; the primer sequences are available in Appendix A.

### 2.9. Transmission Electron Microscopy (TEM)

Sample preparation of A498 cells for transmission electron microscopy was performed as described by Graham, et al. [33]. Briefly, the drug-treated A498 cell line was fixed with 2% glutaraldehyde for 24 h and 4% paraformaldehyde for over 2 h at room temperature. The mitochondria were observed under a transmission electron microscope (JEOL, #JEM-1010, Akishima, Tokyo, Japan).

### 2.10. Immunocytochemistry (ICC)

Cells were seeded in 12-well plates containing poly D lysine-coated coverslips, and treated with metformin and everolimus for 24 h. The coverslips, to which the drug-treated cells were attached, were washed with PBS and fixed with 4% paraformaldehyde solution at room temperature for 30 min. The cells were incubated with 0.1% Triton X-100 at room temperature for 5 min and then blocked with 1% goat serum/PBS for 30 min at room temperature. The cells were then incubated for 30 min by dilution of the primary antibody in 1% goat serum, washed with PBS, and incubated with FITC-conjugated secondary antibody for 30 min. Finally, after adding DAPI for nuclear staining, the cells were washed and mounted. The results were captured using a fluorescence microscope at 100× magnification.

### 2.11. LysoTracker and LysoSensor Staining

The vital mitochondrial and lysosomal dyes LysoTracker Deep Red (Invitrogen, #L12492, Waltham, MA, USA) and LysoSensor Green DND-189 (Invitrogen, #L7535) were diluted in water to final concentrations of 1 µM and 50 nM, respectively. Each dye was then added to the cell culture medium and incubated at 37 °C in the dark for 15 min. The cells were then rinsed twice in fresh culture medium before observation by fluorescence microscopy.

### 2.12. JC-1 Staining

The cells were seeded in an 8-well chamber slide (Merck, #C7182, Kenilworth, NJ, USA) and cultured at 37 °C in a 5% CO_2_ incubator overnight. After the supernatant was removed, JC-1 (Sigma–Aldrich, #420220) working solution (4 μmol/L, 100 μL) was added, and the cells were cultured at 37 °C in the dark for 30 min. The cells were washed twice with 200 μL of PBS, and observed under a fluorescence microscope.

### 2.13. Xenograft

BALB/c nude mice (female, 4 weeks old, 20 g) were purchased from Orient Bio, and maintained under pathogen-free conditions. A498 cells (5 × 10^6^ cells per 0.1 mL Hank’s Balanced Salt Solution) were injected subcutaneously into the left flank. Metformin (150 mg/kg) and everolimus (10 mg/kg) were administered intraperitoneally three times per week for four weeks. The tumor sizes were measured every 2–3 days using a digital caliper, and the tumor volumes were calculated using the formula volume = π/6 (length × width^2^). The animal study protocol was reviewed and approved by the Institutional Animal Care and Use Committee of Hanyang University (2019-0177A).

### 2.14. ATP Measurement

The intracellular ATP concentration was measured using an EZ ATP assay kit (DoGenBio, #DG-ATP100). Fresh cells were lysed using an ATP assay buffer, and immediately subjected to deproteinization. The deproteinized lysate was then added to the reaction mixture. After incubation for 30 min at room temperature in the dark, absorbance was measured at 570 nm using a microplate reader.

### 2.15. Immunohistochemistry (IHC)

Paraffin-embedded xenograft tissues were first cut into sections less than 3 mm thick, to prepare the tissue slides. The tissue slides were deparaffinized and hydrated using xylene and alcohol, washed in 0.6% H_2_O_2_/methanol, and treated with 0.1% Triton-X 100. After washing three times with PBS, blocking with 10% goat serum was performed to suppress nonspecific responses. To examine the expression of p62 (Invitrogen, #PA5-27247) and LC3B (Invitrogen, #PA5-32254), the slides were incubated with a corresponding primary antibody overnight at 4 °C; the next day, they were incubated with a secondary antibody at room temperature. The slide was then reacted with the DAB kit (Vector Laboratories), in accordance with the manufacturer’s instructions. The slides were counterstained with hematoxylin-1 and a bluing solution. The slides were observed under an optical microscope (Leica DM 4000B microscope, Leica Microsystems Inc., Buffalo Grove, IL, USA) at 200× magnification, and images were captured using a digital camera (DFC310 FX, Leica Microsystems Inc., Buffalo Grove, IL, USA). For each image, the area of stained brown color was quantified as a percentage of the whole tissue area (LAS V4. 1.0; Leica).

### 2.16. Statistical Analysis

The results are expressed as the mean ± SEM. All experiments were performed at least three times, and all samples were analyzed in triplicate. Most statistical comparisons were performed by one-way ANOVA, followed by Bonferroni’s post hoc test using Prism 7 (GraphPad, San Diego, CA, USA) to compare the groups. Statistical significance was set at *p* < 0.05.

## 3. Results

### 3.1. Metformin and Everolimus Inhibit Cell Viability in the Caki-1, A498, and ACHN Cell Lines

To set the proper drug concentration for combination, we checked the concentration range through various studies [20,34,35,36,37,38], and evaluated the dose-related cytotoxicity of metformin and everolimus to initiate the synergic effects. The Caki-1, A498, and ACHN cell lines were treated with various concentrations of metformin (0, 1, 10, 20, and 50 mM) for 24 h. In the three cell lines, a significant change in cell viability was observed with increasing metformin concentrations (Appendix A). After treatment with the control (absence) and everolimus (1, 5, 10, and 20 μM) for 24 h, the three cell lines showed a significant dose-dependent decrease in cell viability. Everolimus also showed a significant dose-dependent decrease in cell viability (Appendix A). Based on the cell viability results for each drug treatment, we investigated whether the combination of metformin and everolimus had a synergistic effect on RCC inhibition. In the control, metformin, everolimus, and combination treatment, the combination treatment (metformin, 20 mM; everolimus, 10 μM) resulted in the lowest cell viability in all three cell lines (metformin, everolimus, and metformin + everolimus: 66.9, 47.5, and 34.1% for Caki-1; 43.1, 45.4, and 25.6% for A498; and 56.8, 62.7, and 29.6% for ACHN, respectively) (Figure 1A).

To study the impact of the treatment options on the induction of apoptosis, Caki-1, A498, and ACHN cells treated with the control (absence), metformin, everolimus, and the combination of metformin and everolimus, were examined by flow cytometric analysis. For all cell lines, treatment with the combination of drugs demonstrated a tendency for extensive apoptosis compared to the control or each treatment group (control, metformin, everolimus, metformin + everolimus: 2.4, 3.6, 3.2, and 3.9% for Caki-1; 2.9, 5.7, 3.5, and 6.7% for A498; and 0.1, 2.4, 1.5, and 4.4% for ACHN, respectively) (Figure 1B). The synergistic inhibitory effect of the combination treatment on the downstream target proteins of the mTOR signaling pathway (p-p70S6K and p-4EBP1) was examined via Western blot analysis. The combination treatment decreased the activation of mTOR signaling members, compared with everolimus or metformin treatment alone, in all cell lines (Appendix A). Thus, we found that the combination treatment inhibited the mTOR signaling pathway in the RCC cells.

### 3.2. Combination Treatment Suppresses Cell Growth and Migration of RCC

To evaluate the effects of the treatment options on cell migration and invasion, we quantified the wound healing and invasion assays. In the wound healing assay and the invasion assay, treatment with metformin or everolimus alone inhibited the wound healing and invasive capacity of Caki-1, A498, and ACHN cells compared to the control group (Figure 1C,D and Appendix A). The combination treatment showed the most effective inhibition of cell migration and invasion in all cell types. These data indicated that the combination treatment synergistically inhibited cancer cell migration and invasion.

To confirm whether metformin presented a synergic effect with everolimus on cancer progression in vivo, we used a xenograft model injected with A498 cells. In the xenograft model, metformin (150 mg/kg) and everolimus (10 mg/kg) were administered. We measured the tumor size three times per week, from the time the tumor size reached 500 mm^3^ to 4 weeks after each treatment option. The combination treatment showed the lowest tumor weight and size among the treatment options, followed by everolimus, metformin, and the control (Figure 1E–G). Although the statistical difference in outcomes was not significant (Figure 1F; everolimus, *p* = 0.072) due to the small sample size, the combination treatment was shown to synergistically reduce the tumor weight and size.

### 3.3. Association of the Combination Treatment with Mitochondrial Transporters and Mitophagy

To delineate the mechanism underlying the synergistic effects of the combined treatment, we analyzed gene expression microarrays of A498 cells treated with metformin and everolimus. Significant differential gene expression was detected for 2079 genes, including 1287 upregulated and 792 downregulated genes, under the combined treatment with everolimus and metformin [*p*-value with a false discovery rate (FDR) < 0.05, |fold change (FC) of the combination treatment per control| > |FC of the everolimus treatment per control or FC of the metformin treatment per control|, and |FC of the combination treatment per control| ≥ 1.5] (Figure 2A). We evaluated changes in gene expression according to the treatment options using Gene Ontology (GO) and Kyoto Encyclopedia of Genes and Genomes (KEGG) pathway analyses. The GO functional analysis showed that significantly regulated genes were associated with multiple cancer-related categories, including biological processes, cellular components, and molecular functions (Appendix A), while the KEGG pathway analysis revealed that the various clusters, including metabolic pathways, pathways in cancer, and lysosomes, were changed by the combined treatment, suggesting that the combination of metformin and everolimus is closely related to cancer cell maintenance (Appendix A). In the analysis to determine a more detailed mechanism, significant changes in mitochondria-related genes were observed, including four mitochondrial transporter-related genes (SLC25A15, SLC25A22, SLC25A30, and SLC25A46) and two mitophagy marker genes (PINK1 and OPTN). Mitochondrial transporter genes were generally scarce, while the mRNA levels of mitophagy genes were generally more abundant in the combination treatment than for mono-treatment (Figure 2B,C). These results show that the combination treatment damaged mitochondrial transporters and enhanced mitophagy activity.

### 3.4. Combination Treatment Modulates the Mitochondria Fusion-Fission Cycle

Because mitophagy occurs after mitochondrial fission, we hypothesized that the combination treatment might also affect mitochondrial dynamics. To identify the effect of the combination treatment on mitochondrial dynamics, we examined mitochondrial fusion-fission cycle-related genes via microarray analysis. Representative mitochondrial fusion-fission-related genes (fission: MIEF2, DNM1L, and FIS1; fusion: OPA1, MFN1, and MFN2) were selected. By the combination treatment, the mRNA levels of the fission genes were significantly increased, and those of the fusion genes were decreased (Figure 2D).

Disruptions to mitochondrial dynamics have a complicated impact on resistance to various types of stress [39]. Inhibition of stress resistance originating from the disruption of mitochondrial fusion genes has been suggested as a target for therapy in diverse tumor types [39]. We hypothesized that RCC could be resistant to conventional treatment via a mutation in mitochondrial dynamics-related genes. To prove our hypothesis, we analyzed the genetic mutations of six mitochondrial fusion-fission-related genes from The Cancer Genome Atlas Kidney Renal Clear Cell Carcinoma (TCGA-KIRC) database for clear cell RCC. Of these genes, the fusion marker genes (OPA1 and MFN1) showed a higher level of gene amplification than other genes (Figure 3A). Additionally, we examined the mRNA levels of mitochondrial biosynthesis-related genes (NRF1 and TFAM), fusion marker genes (OPA1, MFN1, and MFN2), and fission marker genes (MIEF2, DRP1, and FIS1) by the drug treatment. Mitochondrial biosynthesis-related genes and fission marker genes were downregulated, and fusion marker genes were upregulated, in the combination treatment, compared to the control and separate drug treatments in all cell lines (Figure 3B–D). We observed the effects of metformin and everolimus on mitochondria using transmission electron microscopy. In the control group, many long-shaped mitochondria were observed, while in the metformin or everolimus mono-treatment group, the mitochondria were shortened and the number was reduced. Furthermore, massive mitophagy was observed after treatment with the combination of metformin and everolimus (Figure 3E). These results showed that the combination treatment enhanced drug therapeutic effects by decreasing mitochondrial fusion and increasing mitochondrial fission, suggesting that the mitochondrial morphology shifted toward a fragmented network.

### 3.5. Impact of the Combination Treatment on Mitochondrial Dynamics

To confirm whether the treatments applied to the Caki-1, A498, and ACHN cell lines activated the mitophagy, we examined the levels of autophagy-related markers, including p62 and LC3B, by immunocytochemistry. In an assessment of the molecular markers of mitochondrial dynamics in A498, many more foci with higher enrichment of LC3B were found in cells treated with the drug combination compared to those treated with metformin alone or everolimus alone (Figure 4A). Compared with the LC3B results, fewer foci with lower enrichment of p62 were found in cells treated with the combination treatment. Additionally, we examined whether the damaged mitochondria were degraded by fusion with lysosomes using LysoTracker and LysoSensor staining. Higher enrichment was found in the combination treatment compared to metformin or everolimus alone (Figure 4B). To identify the effects of damaged mitochondria in which mitochondrial transporter-related genes were downregulated, a fluorometric analysis, after staining with the fluorescent dye JC-1, was performed (Figure 4C). Everolimus alone significantly increased the green fluorescence of the JC-1 monomers. The greatest number of damaged mitochondria and the highest green fluorescence intensity were observed in the combination treatment group. The immunocytochemistry, LysoTracker, LysoSensor, and JC-1 staining results for Caki-1 and ACHN were similar to the results for A498 (Appendix A).

### 3.6. Synergic Effect of the Combination Treatment on the Mitochondrial Fusion-Fission Cycle in RCC

To reveal the drug-induced damage to mitochondria, which are known as the hubs of energy production, we measured the intracellular ATP concentration. The relative ATP concentration in the combination treatment was significantly lower than that in the everolimus treatment in the Caki-1, A498, and ACHN cell lines, indicating an overall worsening of mitochondrial function (metformin, everolimus, and metformin + everolimus: 84.7%, 51.0%, and 28.6% in Caki-1; 76.7%,52.7%, and 25.1% in A498; and 87.9%, 60.0%, and 42.9% in ACHN, respectively) (Figure 5A). Based on our results, showing that the combination treatment synergistically enhanced anti-cancer effects in vitro, we conducted a histological evaluation in vivo. We measured the mRNA levels of four mitochondrial transporter-related genes (SLC25A15, SLC25A22, SLC25A30, and SLC25A46) and two mitophagy-related genes (PINK1 and OPTN) in xenograft tissue using qRT-PCR, which confirmed the results in vitro (Figure 5B).

To evaluate the disruption of mitochondrial dynamics in vitro, we examined the levels of autophagy-related markers, including p62 and LC3B, using immunocytochemistry. In xenografts with A498, higher expression of LC3B and lower expression of p62 were found with the combination treatment, compared to the other treatment options (Figure 5C). KEGG analysis was performed to identify potential pathways and reveal their function in RCC, with respect to the treatment options (Figure 5D); calcium signaling, apelin signaling, cGMP-PKG signaling, MAPK signaling, and Hippo signaling have been indicated. As shown in the bar chart, the combination treatment can control the calcium signaling pathway, which regulates mitochondrial effectors.

## 4. Discussion

In this study, we investigated the underlying mechanisms for the synergic effect of the combination treatment with metformin and everolimus. Through microarray analysis, we revealed that the combination treatment modulated mitochondrial dynamics, and ultimately caused a massive mitophagy, and this was confirmed by xenograft tissues and TEM (Figure 6). Our results demonstrated that the combination treatment can synergistically enhance the anti-cancer effect on RCC by inducing mitochondrial dysfunction.

Metformin inhibits the in vitro proliferation, distant invasion, and migration of RCC. In the Caki-1, A498, and ACHN cells, metformin had an inhibitory effect on proliferation in a concentration-dependent manner at values up to 10 mM, which was similar to the results of previous studies with Caki-1, Caki-2, or 786-O cells [20,40]. Additionally, our study showed that metformin decreased the migration and wound-healing ability of A498, Caki-1, and ACHN cells. These results suggest that metformin has important potential effects on tumor suppression, and could inhibit distant invasion and migration of RCC. Furthermore, metformin induced apoptosis of Caki-1, A498, and ACHN cells. The combination treatment with metformin and everolimus showed a synergic effect on the proliferation, migration, invasion, and apoptosis of Caki-1, A498, and ACHN cells compared to each treatment alone. Therefore, these results suggest that metformin might help enhance the lethal effects of traditional chemotherapy drugs toward RCC.

In the synergic effect of the combination treatment in vitro, differences in the treatment response were observed among the Caki-1, A498, and ACHN cells (Figure 1). Concentration-dependent responses to metformin alone were well observed in the A498 cells, followed by the ACHN and Caki-1 cells. Caki-1 was less sensitive, indicating a difference in sensitivity toward the action of metformin among the cell lines, as explained by the results of previous studies [20,41]. Among the concentration-dependent responses to everolimus alone, the relative cell viability showed greater differences in the A498 cells than the Caki-1 and ACHN cells. These results indicate the differential sensitivity of the three cell lines toward metformin, everolimus, or the combination treatment.

Based on the results of the effects of the combination treatment on the mTOR pathway, we aimed to identify the direct target genes related to treatment responses. Based on the results of the microarray analysis, we found that the combination treatment regulated mitochondrial transporter-related genes (SLC25A15, SLC25A22, SLC25A30, and SLC25A46) and mitophagy marker genes (PINK1 and OPTN), which is consistent with the results of the qRT-PCR analysis. Of the six genes, SLC25 is the largest solute transporter family in the human mitochondrial carrier family [42]. The main role of SLC25 is to transport solutes across the impermeable inner membrane of mitochondria for important cellular processes, such as iron sulfur cluster and heme synthesis, heat production, amino acid catabolism and interconversion, macromolecular synthesis, and fat and sugar oxidative phosphorylation [43,44]. Some mutations in the SLC25A15 gene were related to human hyperornithinemia, hyperammonemia, and homocitrullinuria syndrome because the ornithine cycle was disrupted. SLC25A25 plays a role in the net uptake or efflux of adenine nucleotides into or from the mitochondria as an ATP-Mg/Pi carrier that mediates Mg-ATP transport in exchange for phosphate [45].

To accurately identify the mechanisms underlying the effects of the combination treatment on mitochondria damage following the regulation of mitochondrial transporter genes and mitophagy genes, we investigated mitochondrial dynamics genes. In two fusion genes among six mitochondrial fusion-fission-related genes selected from microarray analysis, the presence of genetic mutations was identified using TCGA-KIRC data. The results indicated that the combination treatment enhanced the treatment response by decreasing mitochondrial fusion and increasing mitochondrial fission and mitophagy.

In conclusion, we demonstrated that the combination of metformin and everolimus inhibits the mitochondrial functions by inducing mitochondrial damage and activating excessive mitochondrial fission and mitophagy in RCC. Our study suggests that increasing sensitivity to conventional drugs will facilitate the development of novel therapeutic strategies for refractory cancers with frequent metastases, such as RCC.

## Figures and Tables

**Figure 1 genes-13-01211-f001:**
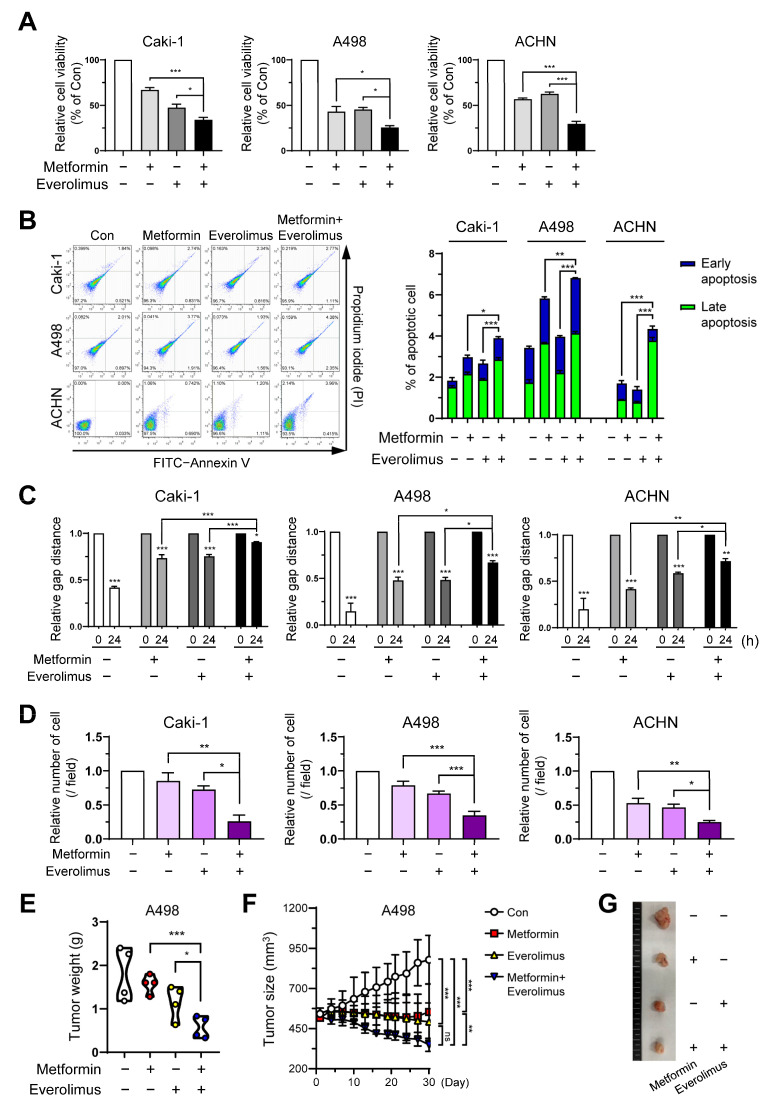
Combination treatment of metformin and everolimus synergistically inhibits RCC: (**A**) Caki-1, A498, and ACHN cells were treated with the control (absence) and metformin (20 mM), and everolimus (10 μM). Fluorescence values for cell viability measurements were normalized to the controls and expressed as a percentage of control; (**B**) Cells were incubated in the control (absence), with metformin (20 mM) and everolimus (10 μM) separately, and with a combination of the drugs for 24 h, and then apoptosis was assessed using the annexin V-FITC apoptosis detection kit, followed by flow cytometry analysis; (**C**) Wound-healing assay at 0 and 24 h in the treatments with the control, metformin (20 mM), everolimus (10 μM), and combination of drugs. Percentage of gap distance at 0 h of control group; (**D**) Quantification graph for the invasive ability of cells treated with control, metformin, everolimus, and drug combination; (**E**) Graphs representing the tumor weight of A498 cell xenografts (n = 4, per group) treated with the control, metformin (150 mg/kg), everolimus (10 mg/kg), and combination of the drugs; (**F**) Graphs representing the average tumor sizes of xenografts according to the treatment options; (**G**) Pictures of the excised tumors of the control and treatment groups. Data are presented as the mean ± standard error of the mean. * *p* < 0.05; ** *p* < 0.01; *** *p* < 0.001; and ns *p* > 0.05 vs. control.

**Figure 2 genes-13-01211-f002:**
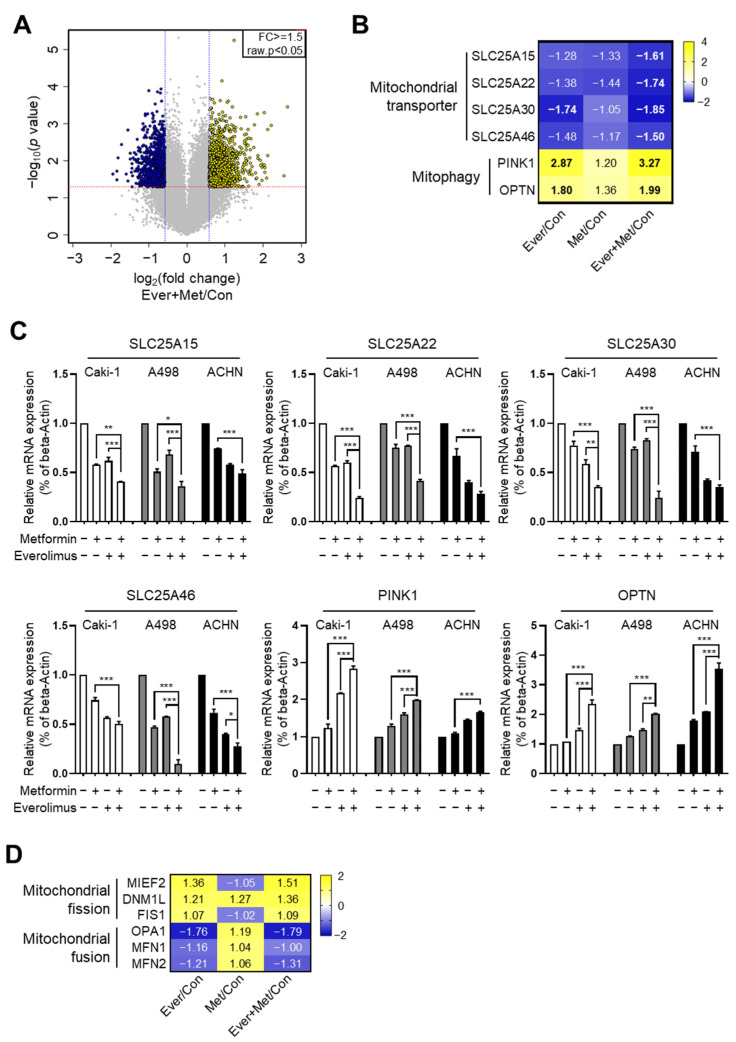
Synergistic effects of the combination treatment of metformin and everolimus induced mitochondrial dysfunction: (**A**) Volcano plot illustration of differentially expressed genes under the combination treatment; (**B**) Heat map visualization of six differentially regulated genes related to mitochondrial transporters and mitophagy; (**C**) qRT-PCR for SLC25A15, SLC25A22, SLC25A30, SLC25A46, PINK1, and OPTN treated with the control, metformin, everolimus, and combination of the drugs in Caki-1, A498, and ACHN cells. Data are presented as the mean ± standard error of the mean. * *p* < 0.05; ** *p* < 0.01; and *** *p* < 0.001 vs. combination treatment group of metformin and everolimus; (**D**) Heat map visualization of six differentially regulated genes related to mitochondrial fusion-fission.

**Figure 3 genes-13-01211-f003:**
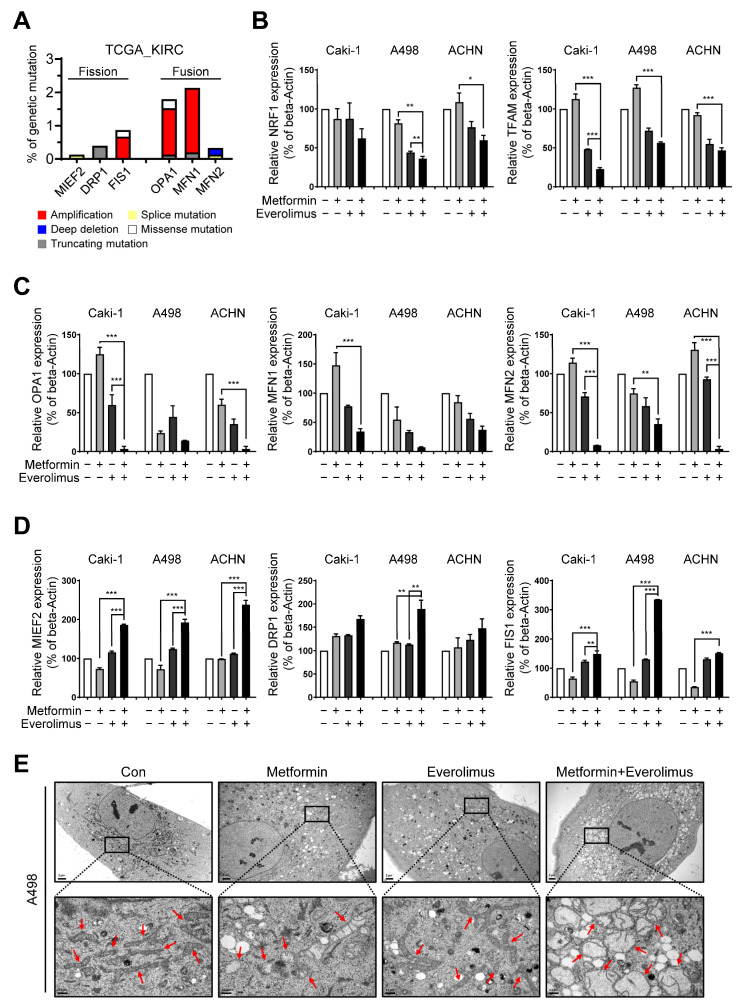
Combination treatment regulates mitochondrial dynamics: (**A**) Several mitochondrial fusion-fission-related genes presented the amplification mutation based on the TCGA-KIRC data; (**B**–**D**) qRT-PCR for (**B**) mitochondrial biosynthesis-related genes, (**C**) fusion marker genes, and (**D**) fission marker genes in the Caki-1, A498, and ACHN cells treated with the control, metformin, everolimus, and drug combination; (**E**) Representative TEM images of mitochondria (indicated with red arrows) in drug-treated A498 cells. Bar: 2 μm (upper) and 0.5 μm (bottom). Data are presented as the mean ± standard error of the mean. * *p* < 0.05; ** *p* < 0.01; and *** *p* < 0.001 vs. combination treatment group of metformin and everolimus.

**Figure 4 genes-13-01211-f004:**
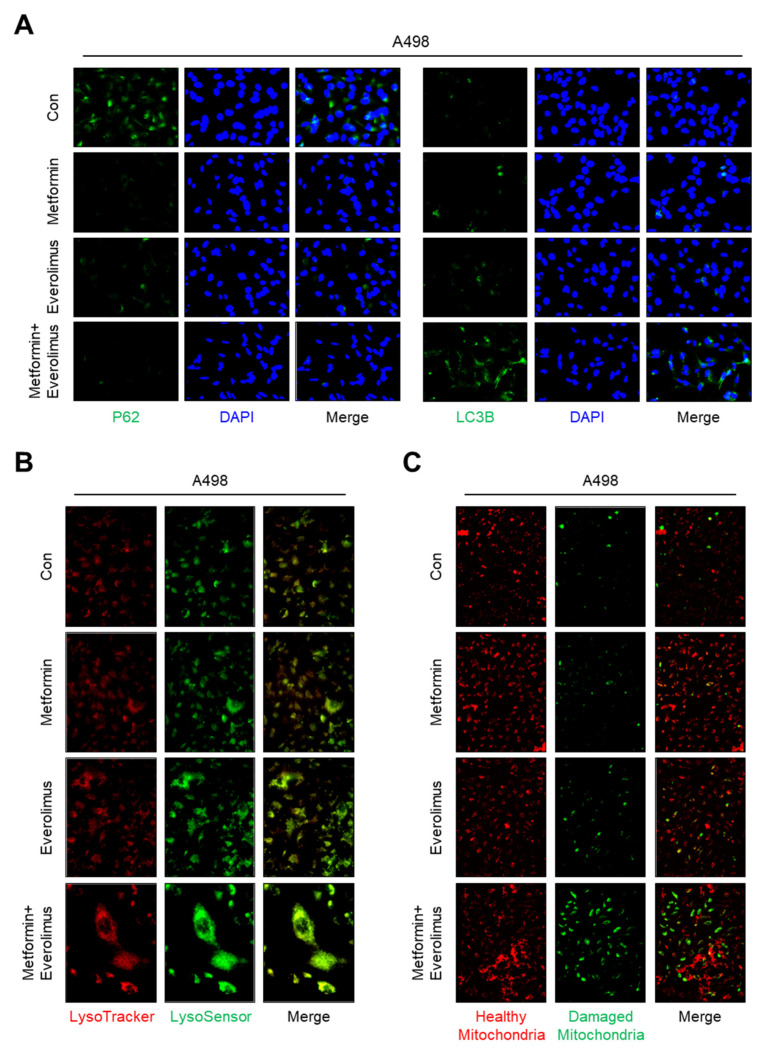
Combination treatment of metformin and everolimus has synergic effects that lead to mitophagy following mitochondrial damage: (**A**) Representative images of immunocytochemistry analysis with anti-P62 (green), anti-LC3B (green), and DAPI (blue); (**B**) Cells were incubated with the drug for 24 h, followed by staining with LysoSensor (green) and Lysotracker (red); (**C**) Fluorescence images of cells stained with JC-1 after treatment according to the drug options (red; healthy mitochondria and green; damaged mitochondria).

**Figure 5 genes-13-01211-f005:**
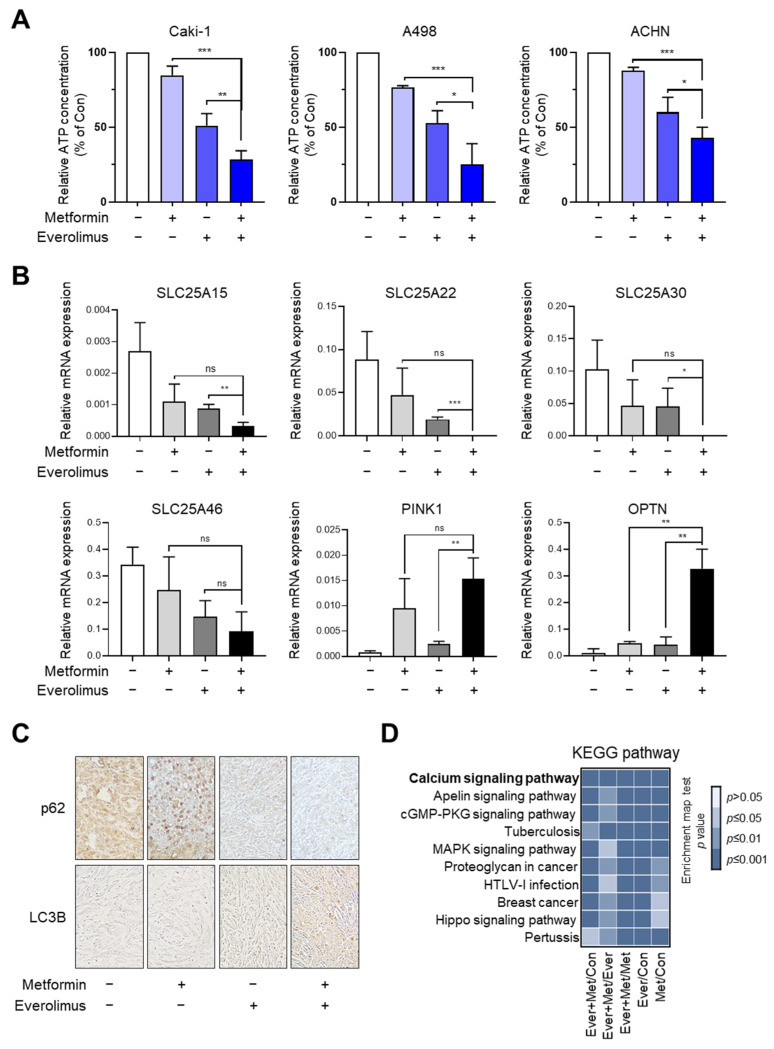
Combination treatment of metformin and everolimus synergistically reduces RCC growth by disrupting mitochondrial function: (**A**) Relative ATP concentration in Caki-1, A498, and ACHN xenografts treated with the control, metformin, everolimus, and combination of the drugs; (**B**) qRT-PCR for mitochondrial transporter and mitophagy-related genes in the tissue of xenografts treated with the control, metformin, everolimus, and combination of drugs; (**C**) Immunohistochemistry images of p62 and LC3B in the tissue of xenografts treated with the control, metformin, everolimus, and combination of drugs; (**D**) Bar chart showing the KEGG enrichment pathways in the mitochondrial dynamics marker genes. Data are presented as the mean ± standard error of the mean. * *p* < 0.05; ** *p* < 0.01; *** *p* < 0.001; and ns *p* > 0.05 vs. combination treatment group of metformin and everolimus.

**Figure 6 genes-13-01211-f006:**
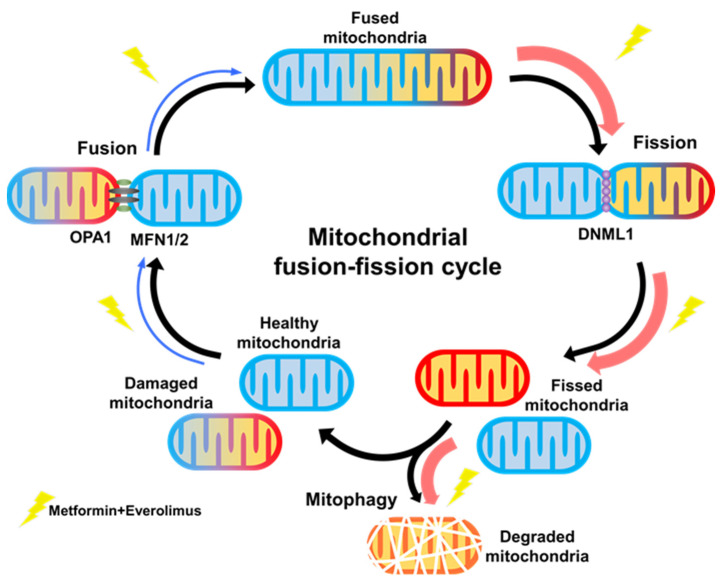
Graphical abstract.

## Data Availability

All study data are included in the article and Appendix A.

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
