# Peer review of "Synergic Effect of Metformin and Everolimus on Mitochondrial Dynamics of Renal Cell Carcinoma"

_genes, 2022, doi:10.3390/genes13071211_

Round 1

Reviewer 1 Report

The paper by Yoon and team members showed that the combination of everolimus and metformin inhibited RCC growth by disrupting mitochondrial dynamics. The study seems to be interesting and worth publishing. The methods section is clearly described and results are well discussed. The results are convincing to support the conclusion.

Minor comments

The authors may add more details in the introduction section. For example, add details about mitochondrial dynamics in various disorders from the literature.

Author Response

Reviewer 1

The paper by Yoon and team members showed that the combination of everolimus and metformin inhibited RCC growth by disrupting mitochondrial dynamics. The study seems to be interesting and worth publishing. The methods section is clearly described and results are well discussed. The results are convincing to support the conclusion.

Minor comments

The authors may add more details in the introduction section. For example, add details about mitochondrial dynamics in various disorders from the literature.

- We appreciate reviewer 1’s comments. As suggested by reviewer 1, we added sentences and references about the mitochondrial dynamics in malignant tumors.

Reviewer 2 Report

Lee et al. have done a good work to unravel the synergistic effect of metformin and everolimus in renal cell carcinoma. They demonstrated that combination of everolimus and metformin inhibited renal cell carcinoma growth by disrupting mitochondrial dynamics and confirmed the synergistic effect in vivo. However, there are lots of typographical errors. The manuscript may be accepted after addressing the following comments.

There should be a space between number and unit. For example, in line 173, 178 and 180, 37°C should be written as 37 °C. The authors should check others.

CO2 and H2O2 should be written as CO2 and H2O2

In line 184, A498 cells (5×106 cells per 0.1 mL Hank’s Balanced Salt Solution)…

-Does it mean 5×106 cells per 0.1 mL?

In page 5, The animal study protocol was reviewed and approved by the Institutional Animal Care and Use Committee of Hanyang University. The authors should mention the approved protocol number.

In line 214, the authors should delete the word “Most”. What was the other software used for statistical analysis?

In Figure 1B, the authors should include the original figure of FACS analysis showing apoptotic cell distribution beside the bar diagram. Statistical difference should be included in the bar diagram.

Figure S1 showing in vitro scratch migration assay is not clear. The authors should improve the figure, better to repeat the migration assay to capture improved image.  

Round 2

Reviewer 3 Report

-